# Antibacterial Activity of Synthetic Cationic Iron Porphyrins

**DOI:** 10.3390/antiox9100972

**Published:** 2020-10-10

**Authors:** Artak Tovmasyan, Ines Batinic-Haberle, Ludmil Benov

**Affiliations:** 1Division of Neurobiology, Ivy Brain Tumor Center, Barrow Neurological Institute, Phoenix, AZ 85013, USA; artak.tovmasyan@barrowneuro.org; 2Department of Radiation Oncology, Duke University School of Medicine, Durham, NC 27710, USA; ibatinic@duke.edu; 3Department of Biochemistry, Faculty of Medicine, Kuwait University, Kuwait City 13110, Kuwait

**Keywords:** antibacterial, iron porphyrin, antibiotic resistance, bactericidal, cationic metalloporphyrin

## Abstract

Widespread antibiotic resistance demands new strategies for fighting infections. Porphyrin-based compounds were long ago introduced as photosensitizers for photodynamic therapy, but light-independent antimicrobial activity of such compounds has not been systematically explored. The results of this study demonstrate that synthetic cationic amphiphilic iron *N*-alkylpyridylporphyrins exert strong bactericidal action at concentrations as low as 5 μM. Iron porphyrin, FeTnHex-2-PyP, which is well tolerated by laboratory animals, efficiently killed Gram-negative and Gram-positive microorganisms. Its bactericidal activity was oxygen-independent and was controlled by the lipophilicity and accumulation of the compound in bacterial cells. Such behavior is in contrast with the anionic gallium protoporphyrin IX, whose efficacy depends on cellular heme uptake systems. Under aerobic conditions, however, the activity of FeTnHex-2-PyP was limited by its destruction due to redox-cycling. Neither iron released from the Fe-porphyrin nor other decomposition products were the cause of the bactericidal activity. FeTnHex-2-PyP was as efficient against antibiotic-sensitive *E. coli* and *S. aureus* as against their antibiotic-resistant counterparts. Our data demonstrate that development of amphiphilic, positively charged metalloporphyrins might be a promising approach in the introduction of new weapons against antibiotic-resistant strains.

## 1. Introduction

In a recent report, the WHO pointed to antimicrobial resistance as a global problem that poses a threat for the management of diseases caused by viruses, bacteria, and fungi [1]. The number of antibiotic-resistant bacterial strains is increasing all over the world, which potentially may lead to a point when infections would become untreatable with currently available drugs. The routine answer to this problem has been the introduction of new antibiotics that overcome bacterial resistance. In recent years, however, a drop in the research and introduction of new antibiotics has been observed [2,3,4]. Patients with infections due to antimicrobial-resistant organisms cost the health care system much more than patients infected with antimicrobial-susceptible pathogens [5]. Two species, Gram-positive *Staphylococcus aureus* and Gram-negative *Escherichia coli* appeared to be among the most common cause of antibiotic-resistant infections [6].

The light-independent, reviewed in [7], and photodynamic antimicrobial activities of metalloporphyrins [8,9,10] have attracted interest as potential alternatives to existing antibiotics [11,12,13]. Non-iron metalloporphyrins have been reported to act as efficient light-independent microbicides against Gram-positive, Gram-negative and mycobacteria, with gallium protoporphyrin IX (GaPPIX) being the most efficient [14,15]. Protoporphyrin IX-based metalloporphyrins are anionic and enter the cell through the heme-uptake system of bacteria. Consequently, species that do not express heme uptake systems are resistant [7,14]. The exact mechanism of metalloporphyrins’ light-independent antibacterial action is not known. Supposedly, it is related to inhibition of some essential metabolic pathways, depending on heme-containing enzymes/proteins [7]. On the other hand, the higher sensitivity of catalase- and SOD-deficient mutants, and the fact that anaerobically grown cultures were resistant to non-iron metalloporphyrins, i.e., such compounds are toxic only to actively respiring bacteria [14], suggested that they cause cell damage by either compromising cell respiration or by inducing oxidative stress, or both. Modifications of the peripheral substituents of naturally derived metalloporphyrins dramatically affected their antibacterial activity [14], which stresses the importance of similarity to heme for antibacterial action [7].

A potential advantage of metalloporphyrin-based antibacterials is their low toxicity to eukaryotic cells [7,14]. Various porphyrin-based therapeutics, including manganese porphyrin-based SOD mimics, have shown minimal toxicity to laboratory animals and are now in five phase II human clinical trials [16,17,18,19,20,21,22,23]. In our preliminary studies, we found that cationic Mn porphyrins (MnPs) did not show measurable toxicity to human cell cultures and to mice at concentrations tenfold higher than those killing *E. coli* (unpublished data). While investigating the biological activities of metalloporphyrins, it has been observed that cationic iron *ortho N*-alkylpyridyl porphyrins (FePs) are much more toxic to *E. coli* than their manganese analogs [24,25]. Similar findings were observed by Kawakami’s group against various cancer cells [26]. The difference in biological activity between cationic Fe and Mn porphyrins are likely related to their differential axial coordination. In contrast to MnPs, which have two weakly axially coordinated water molecules, FePs have one strongly coordinated hydroxo axial ligand, and in the trans-position to it, a weakly bound water molecule [24]. The strong axial binding affects FePs’ reactivity in a biological environment as well as pharmacological activities and may in turn induce toxicity to bacteria [24]. The higher toxicity of FePs against *E. coli* and their good tolerability by mammals [24,27] prompted us to further investigate the bactericidal activity of synthetic FePs.

## 2. Materials and Methods

### 2.1. Metalloporphyrins

The metalloporphyrins used in this study (Figure 1) include Fe(III) *N*-alkylpyridylporphyrins, whose synthesis and characterization has been described in detail elsewhere [24]. These FePs were specially designed by maintaining the tetrapyrrole core, while modifying the periphery by attaching aliphatic chains of varying length. Consequently, compounds with desired lipophilicity and three-dimensional structure were obtained. The length of the aliphatic chains attached to the pyridyl nitrogen at the *meso* position varied from ethyl FeTE-2-PyP, Fe(III) *meso-tetrakis*(*N*-ethylpyridinium-2-yl)porphyrin), to hexyl FeTnHex-2-PyP, Fe(III) *meso*-tetrakis(*N*-hexylpyridinium-2-yl)porphyrin) to octyl FeTnOct-2-PyP, Fe(III) *meso*-tetrakis(*N*-octylpyridinium-2-yl)porphyrin). All tested FePs are water-soluble. The charges are omitted throughout the manuscript for clarity. Commercially available gallium protoporphyrin IX (GaPPIX) and hemin were used without further purification. Although isolated as pentachloride salts in solid state, in aqueous systems at physiological pH, cationic FePs coordinate axially with one hydroxo and one water ligands [24], as indicated in Figure 1; for simplicity, axial coordination is omitted throughout the text and in other figures.

### 2.2. Strains and Growth Conditions

The Gram-negative strains used to study the antibacterial activity of metalloporphyrins include: antibiotic-sensitive *Escherichia coli* strain GC4468 (F^−^ Δlac U169 *rpsL*), QC1799 (same as GC4468 plus *Δ sodA3*, *Δ sodB*-kan) provided by Dr. D. Touati [28]; AB1157 [F^−^
*thr-1 leuB6 proA2 his-4 thi-1 argE2 lacY1 galK2 rpsL surE44 ara-14 xyl-15 mtl-1 tsx-33*]; KK204 as AB1157 plus *fur::kan* [29]; MG1655 F^−^ wild-type; LC106, as MG1655 plus *ΔahpF::kan Δ(katG17::*Tn*10)1 Δ(katE12::*Tn *10)1* [30] provided by Dr. J. Imlay, and a clinical *E. coli* isolate resistant to carbapenems provided by Dr. M. John Albert (Faculty of Medicine, Kuwait University). Gram-positive strains: antibiotic-sensitive *Staphylococcus aureus* strain ATCC25923 [31], and antibiotic-resistant clinical isolate CC22-SCCmec IV (provided by Dr. E. Udo, Faculty of Medicine, Kuwait University).

Cultures were grown overnight in Luria Bertani (LB) medium with antibiotics added where necessary. For the preparation of LB plates, 15 g of agar was added to 1 L of liquid LB medium. Working cultures were grown in M9CA medium (M9 salts, 0.2% casamino acids, 0.2% glucose, 3 mg pantothenate, and 5 mg of thiamine per liter) [32]. Growth was monitored by measuring the change of OD at 600 nm using a microplate reader [32]. Viability was assessed by dilution of cultures and plating on LB agar plates for counting colonies. To avoid distortion of results due to variations in rate of growth, whenever necessary, cells were washed and resuspended in PBS containing 0.2% glucose (PBS-glucose).

To avoid light-induced cell damage by metal-free ligands with photosensitizing properties, produced by demetallation of the FePs, cultures and samples were always protected from light.

To test the contribution of oxygen and reactive species derived from it to FePs bactericidal action, additional experiments were performed in an anaerobic Coy chamber, maintaining oxygen content below the detection limit. All solutions were thoroughly degassed and equilibrated for 30 min before use.

### 2.3. Uptake and Accumulation of FePs in E. coli

Cellular uptake of FePs was determined as previously described [33]. In brief, mid-log cultures (OD_600nm_ = 0.5–0.7) grown in M9CA medium were washed and resuspended in PBS-glucose to the same density. FePs were then added to a final concentration of 5 µM and cultures were kept on a shaker at 37 °C for one hour. After the completion of the incubation, cells were rapidly washed with ice-cold PBS, and disrupted by French press. Spectra were recorded and the area under the peak at Soret band was calculated. FeP concentration was determined using a standard curve. Protein concentration was estimated by the method of Lowry [34].

### 2.4. Oxygen Consumption by Bacterial Suspensions

Oxygen consumption was measured as previously described [35] using Biological Oxygen Monitor System (YSI 5300A, YSI Inc., Yellow Springs, OH, USA) equipped with a Clark electrode.

Mid-log *E. coli* suspensions in PBS-glucose were incubated 60 min in the dark with FeTnHex-2-PyP at the indicated concentrations. At the end of the incubation period, 3.0 mL aliquots were transferred to Clark electrode chamber and oxygen consumption was recorded.

### 2.5. Data Analysis

Experiments were repeated at least two times, each sample in triplicate. One-way analysis of variance (ANOVA) was performed using SigmaPlot version 11.0, and *p* value < 0.05 was accepted as statistically significant. Data are presented as mean ± SD.

## 3. Results

### 3.1. Effect of FePs on E. coli Proliferation and Viability

The aim of our initial experiments was to determine how differences in lipophilicity of FePs affect *E. coli* proliferation. Figure 2 shows that none of the tested FePs prevented *E. coli* growth at a concentration of 1.0 µM. At this concentration, FeTE-2-PyP induced ~4 h lag of proliferation, but did not affect the growth rate. At 3.0 µM, the most hydrophilic FeP, FeTE-2-PyP, completely prevented cell proliferation, while the amphiphilic hexyl derivative FeTnHex-2-PyP only decreased the rate of growth. No cell proliferation was observed in the presence of 5 µM FeTE-2-PyP, FeTnHex-2-PyP or FeTnOct-2-PyP.

To find out if the effect of the FePs was only bacteriostatic and bacteria remain viable, after completion of the growth experiments (24 h), the content of the wells where no proliferation was observed (5 µM FePs), was evenly spread on LB agar plates for counting colonies. Cell number in wells at zero time was used for comparison. The number of viable cells decreased by about four log units when cultures were treated with FeTE-2-PyP. No colonies were observed in wells treated with FeTnHex-2-PyP or FeTnOct-2-PyP (Figure 3). These results show that the amphiphilic hexyl and octyl derivatives exerts much stronger bactericidal action than the hydrophilic ethyl analog.

Since FeTnHex-2-PyP demonstrated the lowest toxicity in animal experiments [24], it was selected for further investigations.

The results presented in Figure 3 were obtained after *E. coli* was exposed to FePs for 24 h. It is not clear what would be the effect if contact with the compound was much shorter. To answer this question, *E. coli* was grown to mid-log phase (OD_600nm_ = 0.5–0.6), the cells were thoroughly washed, resuspended in PBS-glucose to avoid cell proliferation during incubation, and treated with 5 μM FeTnHex-2-PyP. Figure 4A shows that short treatment (~15 min) kills ~40% of the cells. Exposure for two hours achieved ~95% viability loss, and exposure for four hours killed practically all bacterial cells.

Another variable that might affect FeTnHex-2-PyP bactericidal effect can be the compound to cell number ratio. Experiments demonstrated that at 5 μM FeP, maximal bactericidal efficiency could be achieved if cell number did not exceed ~7 × 10^8^ cells/mL. Five-fold increase in the initial cell number resulted in a ~30% decrease in FeTnHex-2-PyP bactericidal activity (Figure 4B).

### 3.2. Comparison of FeTnHex-2-PyP with GaPPIX and Hemin

Previous studies have revealed that hemin [7] and non-iron porphyrins such as Ga-protoporphyrin IX [14] were efficient against certain bacterial species. A comparison under the selected experimental conditions shows (Figure 5) that neither Ga-protoporphyrin IX nor hemin suppressed *E. coli* proliferation, even when applied at a concentration four-fold higher than that of FeTnHex-2-PyP. This finding is not surprising, because both compounds are negatively charged and our previous studies have demonstrated that only cationic amphiphilic porphyrins easily cross membranes and accumulate to high concentrations in microbial cells [8,36].

### 3.3. Cellular Uptake of Fe-Porphyrins

The data obtained so far suggest that the bactericidal action of FeTnHex-2-PyP depends on the cellular uptake of the compound, leading to suppression of vital biological functions. Incubation of *E. coli* in the presence of FePs of varying lipophilicity demonstrated that the two amphiphilic FePs, which displayed strong bactericidal activity, accumulated to much higher levels in bacterial cells than the hydrophilic, less active FeTE-2-PyP analog (Figure 6). Based on similar experiments performed with Mn-porphyrin [33] and Zn-porphyrin analogues [8,10], it was established that cationic amphiphilic metalloporphyrins penetrate the cell without the need for a carrier.

### 3.4. Effect of FeTnHex-2-PyP on Oxygen Consumption

FeTnHex-2-PyP is a redox-active compound and potent SOD mimic [24,25]. Thus, a possible reason for its toxicity may be interference with processes that require transfer of electrons. Among them are various metabolic redox reactions, and cellular respiration. To test if respiration is affected by FePs, oxygen consumption by *E. coli* treated with FeTnHex-2-PyP was measured. A concentration-dependent suppression of cellular respiration by the FeP was observed (Figure 7A). Irrespective of the fact that incubation with FeP was reduced to 60 min, loss of viability is inevitable, and suppression of respiration could simply reflect a decreased number of viable cells. To account for the number of viable respiring cells, aliquots taken from the Clark electrode chamber at the time of respiration assay were diluted and plated for counting colonies. When O_2_ consumption was normalized by the number of viable cells in the chamber, it appeared that the FeP increased O_2_ consumption (Figure 7B). The effect was concentration-dependent. One micromole of FeTnHex-2-PyP increased O_2_ consumption ~2 fold compared to the non-treated control, 5 μM FeTnHex-2-PyP produced about 5-fold increase, and 10 μM FeTnHex-2-PyP caused a ~10-fold increase in O_2_ consumption compared to the untreated controls.

An explanation for the increase in O_2_ consumption can be found in the reduction of the Fe(III)P by endogenous reductants, among them thiols and ascorbate [37]. Reduced Fe(II)P is rapidly reoxidized, donating an electron to oxygen and generating superoxide anion radicals, whose dismutation results in H_2_O_2_ production [24]. Therefore, the antibacterial activity of FeP can be attributed to redox-cycling, generating cytotoxic reactive species. This implies that FeP’s toxicity should be manifested only aerobically. When the bactericidal activity of 5 μM FeTnHex-2-PyP was assessed under anaerobic conditions, however, 99.54 ± 0.20% of the cells were killed within 2 h of incubation.

### 3.5. Superoxide Radical, Hydrogen Peroxide, and FeP Decomposition

If H_2_O_2_ was the main cause for FeP’s bactericidal effect, then addition of ascorbate would accelerate FeP redox-cycling and, consequently, FeP toxicity, as was reported before for the Mn-porphyrin analogs [38]. Addition of ascorbate, however, had the opposite effect. One mM ascorbate added to the cell culture simultaneously with FeTnHex-2-PyP, completely abolished its bactericidal activity. At the same time, ascorbate blocked the uptake of FeTnHex-2-PyP by the cells (Figure 6, far right bar). It appears therefore that redox-cycling and H_2_O_2_ production lead to extracellular FeP destruction [24] and consequent dismissal of FeP antibacterial activity. The role of H_2_O_2_ in the mechanism of action of the FeP was further tested on a catalase/peroxidase-deficient mutant (LC106). The mutant strain appeared to be as sensitive to FeTnHex-2-PyP toxicity as its parent. Thus, aerobically, 5 μM FeTnHex-2-PyP killed 88.87 ± 3.01% of the parental (MG1655), and 86.98 ± 4.01% of the catalase/peroxidase-deficient (LC106) cells. Similar results were obtained when the two strains were incubated with FeTnHex-2-PyP anaerobically; parental 90.33 ± 3.24% killed, LC106 89.48 ± 4.52% killed. Neither aerobically nor anaerobically were the differences between the sensitivity of parental and catalase/peroxidase-deficient strains statistically significant. This result was supported by the lack of effect of externally added catalase (1000 units/mL) to *E. coli* suspensions treated with FeTnHex-2-PyP. 

No proof for the contribution of superoxide to the toxic action of FeTnHex-2-PyP was found either. When FeP toxicity to parental and SOD-deficient mutant was compared aerobically, no statistically significant difference was found; parental (GC4468) 89.10 ± 3.02% killed, sodA sodB (QC1799) 91.72 ± 2.26% killed.

Reduction of FePs and their reoxidation leads to porphyrin degradation and release of Fe^3+^ [24], which in the reductive cellular environment is immediately reduced to Fe^2+^ [39]. Liberation of iron has been proposed as a cause of heme toxicity [40] and might be among the reasons for FeP’s bactericidal action. Incubation of bacterial cultures with the iron chelator deferoxamine (2 mM), however, did not protect against FeTnHex-2-PyP bactericidal action (97.73 ± 1.26% of the cells were killed in the absence of deferoxamine and 98.36 ± 0.6105% were killed in cultures with deferoxamine). 

In many bacterial species, including *E. coli*, iron uptake is controlled by the ferric uptake regulon, fur. It represses the expression of heme uptake systems when there is an abundance of iron, thus preventing iron overload. A fur-deficient mutant (KK204), however, was as sensitive to FeTnHex-2-PyP as its parent was (96.91 ± 0.85% vs. 97.73 ± 1.52% killed). This result again shows that FeTnHex-2-PyP toxicity does not depend on bacterial heme uptake system. 

### 3.6. Bactericidal Action of FeTnHex-2-PyP against S. aureus

So far all experiments were carried out using a Gram-negative microorganism, *E. coli*. A Gram-positive species, *S. aureus*, was reported to be highly susceptible to heme toxicity [40]. By analogy, it could be expected that *S. aureus* would be more sensitive to FeP toxicity than is *E. coli*. Indeed, in contrast to *E. coli*, more than 93% of *S. aureus* cells were killed at FeTnHex-2-PyP concentrations as low as 1 μM (Figure 8).

### 3.7. Resistance to Antibiotics and FeP Bactericidal Effect

Resistance to antibiotics had no effect on the antibacterial activity of the FeP. An antibiotic-resistant clinical *S. aureus* isolate CC22-SCCmec IV was as sensitive to killing by 5 μM FeTnHex-2-PyP as the antibiotic-sensitive ATCC25923 strain (99.38 ± 0.57% vs. 99.84 ± 0.75% killed). Similar results were obtained when a carbapenems-resistant *E. coli* clinical isolate was tested (antibiotic-sensitive, 95.85 ± 1.06% vs. carbapenems-resistant, 97.30 ± 0.94% killed).

## 4. Discussion

The results obtained in this study show that ortho cationic Fe(III) *N*-alkylpyridylporphyrins display strong bacteriostatic and bactericidal activities. Bactericidal action depended on FePs lipophilicity, which in turn defines the cellular uptake of the compound. Accumulation of FeP in bacterial cells seems to be crucial for bactericidal action. 

FeTnHex-2-PyP increased oxygen consumption by *E. coli* suspensions in a concentration-dependent manner. Such an increase can be attributed to redox-cycling of the FeP between reduced +2 (Fe^II^P) and oxidized +3 (Fe^III^P) states, while donating an electron to oxygen, converting it to superoxide and, consequently, to hydrogen peroxide; both superoxide and peroxide are bound to the Fe center [24,41] (Scheme 1). Among cellular reductants that can reduce metalloporphyrins, ascorbate and thiols [42,43,44] are found at millimolar concentrations in most cells. The reactions are analogous to redox-cycling of heme iron [45].

No proof that O_2_^●^^−^ and H_2_O_2_ contributed to FeTnHex-2-PyP toxicity was found. The antibacterial activity of FeTnHex-2-PyP was not prevented by catalase or anaerobiosis and neither *sodA sod B* nor catalase/peroxidase-deficient mutants showed higher sensitivity to the FeP than their corresponding parents. These results, along with literature evidence, imply that these reactive species remain bound to the FeP, causing its rapid decomposition into free iron and degradation products [24,25,47]. Neither free Fe nor reactive species or porphyrin degradation products appeared to be a cause of FePs’ bactericidal activity.

FeTnHex-2-PyP was more toxic to a Gram-positive species, *S. aureus*, than to the Gram-negative *E. coli*. This implies that structure and permeability of the cell envelope, which determine the cell penetration and accumulation of the FePs, modulate the toxicity of the metalloporphyrin. Resistance to antibiotics did not affect sensitivity to FeTnHex-2-PyP in either of the tested microbial species.

Similar to heme, mechanisms of FePs bacterial toxicity seem to be complex and require detailed investigations. It has been hypothesized [25] that the toxicity of FePs might be due to their high affinity for axial ligation [48]. High cellular protein concentration makes peptides the most likely candidates for binding FeP. By analogy with heme, one can expect that amino acid residues acting as ligands in naturally occurring hemoproteins (His, Tyr, Met, Lys, and Cys) [49,50] would also have high affinity towards synthetic Fe-porphyrins. Ligation of FeP to amino acid side chains can eventually disrupt essential protein functions, which could cause cell death. Binding of FePs to proteins could interfere with enzymatic, cell signaling, and other peptide functions, and could lead to the disruption of the plasma membrane barrier and the suppression of metabolic pathways, etc.

## 5. Conclusions

In conclusion, at relatively low concentrations (1–5 µM), an amphiphilic FeP exhibited potent antibacterial activity, which depended on the uptake and accumulation of the compound into bacterial cells. Similarly to heme, the tested FeP was more toxic to the Gram-positive *S. aureus* than to the Gram-negative *E. coli*. Aerobically, the antibacterial activity of the FeP has been limited by the porphyrin destruction, a consequence of FeP redox cycling. Therefore, the antibacterial activity of the FeP depends on the intactness of the porphyrin structure and does not result from decomposition products. The lower toxicity of FeTnHex-2-PyP to laboratory animals and its strong bactericidal activity give hope that similar compounds with higher stability might in the future be developed as microbicides against drug-resistant pathogens.

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
