# Peer review of "Antibacterial Activity of Synthetic Cationic Iron Porphyrins"

_antioxidants, 2020, doi:10.3390/antiox9100972_

Round 1

Reviewer 1 Report

Tovmasyan et. al., have studied iron chelated porphyrins to induce toxicity in bacteria leading to reduced cell viability. FeP, cationic metalloporphyrin, proved to be more toxic to Gram-positive bacteria in comparison to Gram-negative E. coli.

1. Did the authors investigate if there was a strong interaction between the surface charge on bacteria and cationic compounds leading to the reduction in viability? Generally, cationic compounds (moreso in case of polymers like PEI) are known to have very strong interaction with cell membranes leading to rupture of cell membrane eventually resulting in cell death. Although, we are not dealing with polymers here, cationic porphyrins might still be exerting some influence when they interact with E. coli. Some basic microscopy might be able to shed light on the interaction between E. coli and metalloporphyrins.

2. Continuing on point 1, dechelation of metal (Fe in this case) might be restoring the fluorescence properties of the porphyrins so microscopy might yield very interesting results . It could be further used to generate phototoxicity (photodynamic therapy). Although, these studies could be a part of follow up research projects.

3. Porphyins in general (mTCPP and pyropheophorbide) are known to be hydrophobic in nature. However, the structures shown in figure 1 seem to be soluble in aqueous mixtures. It might be useful to the readers if the authors can include the solubility of these metalloporphyrins.

4. Line 43: Parenthesis ")" seems like a typo - can be deleted

5. Line 77: Seems to be a typo - "details" can be changed to "detail"

Author Response

Response to Reviewer 1

We are grateful for the help in improving the manuscript and for the useful suggestions.

R1

Open Review

(x) I would not like to sign my review report

( ) I would like to sign my review report

English language and style

( ) Extensive editing of English language and style required

( ) Moderate English changes required

(x) English language and style are fine/minor spell check required

( ) I don't feel qualified to judge about the English language and style

Yes        Can be improved               Must be improved             Not applicable

Does the introduction provide sufficient background and include all relevant references?

(x)          ( )           ( )           ( )

Is the research design appropriate?

(x)          ( )           ( )           ( )

Are the methods adequately described?

(x)          ( )           ( )           ( )

Are the results clearly presented?

(x)          ( )           ( )           ( )

Are the conclusions supported by the results?

(x)          ( )           ( )           ( )

Comments and Suggestions for Authors

Tovmasyan et. al., have studied iron chelated porphyrins to induce toxicity in bacteria leading to reduced cell viability. FeP, cationic metalloporphyrin, proved to be more toxic to Gram-positive bacteria in comparison to Gram-negative E. coli.

  1. Did the authors investigate if there was a strong interaction between the surface charge on bacteria and cationic compounds leading to the reduction in viability? Generally, cationic compounds (moreso in case of polymers like PEI) are known to have very strong interaction with cell membranes leading to rupture of cell membrane eventually resulting in cell death. Although, we are not dealing with polymers here, cationic porphyrins might still be exerting some influence when they interact with E. coli. Some basic microscopy might be able to shed light on the interaction between E. coli and metalloporphyrins.

 - Thank you for this idea. The mechanism of antibacterial action of metalloporphyrins is currently under investigation. Metalloporphyrin-membrane interactions will be studied in detail using confocal fluorescent microscopy and electron microscopy. Loss of membrane barrier function will be investigated by propidium iodide staining and flow cytometry. Our previous investigations with cancer cell cultures have shown that amphiphilic Zn-porphyrins disperse in membrane lipid bilayer (Ezzeddine et al. 2013, JBC 288(51): 36579-36588) and that amphiphilic Mn-porphyrins accumulate to high levels in E. coli cell envelope (Kos et al. 2009, J Med Chem 52(23): 7868-7872).

  1. Continuing on point 1, dechelation of metal (Fe in this case) might be restoring the fluorescence properties of the porphyrins so microscopy might yield very interesting results . It could be further used to generate phototoxicity (photodynamic therapy). Although, these studies could be a part of follow up research projects.

- Thank you again for the good suggestions. In addition, if parameters are properly chosen, phototoxicity can help to point the exact place of the porphyrin in the cell. Because singlet oxygen has very short diffusion distance, only structures that are in close proximity to the porphyrin molecule will be damaged. During the oxidative degradation of Fe-porphyrins, however, the Fe center oxidizes the ligand before Fe is released, and degraded ligand may not be fluorescent. We are also planning to use fluorescent analogous metal-free ligands, Zn-porphyrins and In-porphyrins.

  1. Porphyins in general (mTCPP and pyropheophorbide) are known to be hydrophobic in nature. However, the structures shown in figure 1 seem to be soluble in aqueous mixtures. It might be useful to the readers if the authors can include the solubility of these metalloporphyrins.

- All Fe-porphyrins used in this study are water-soluble. A statement has been in included in the manuscript.

  1. Line 43: Parenthesis ")" seems like a typo - can be deleted

- Yes, it was typo. Deleted.

  1. Line 77: Seems to be a typo - "details" can be changed to "detail"

- Yes, it was typo. Deleted.

Reviewer 2 Report

The manuscript describes the light-independent antimicrobial activity of synthetic cationic amphiphilic iron N-alkyl pyridylporphyrins. This is an interesting paper which is doubtlessly suitable for publication in “Antioxidants”. The paper is complete and well structured. The description of experimental procedure is detailed and results critically analyzed and commented. In summary, the measurements are adequate and the derived conclusions are consistent. Particularly interesting are results obtained for FeTnHex-2-Py that presents a strong bactericidal activity and lower toxicity to laboratory animals.

I think the manuscript can be accepted in this form and it needs no revisión.

Author Response

Response to Reviewer 2

Thank you for your positive remarks!

Open Review

(x) I would not like to sign my review report

( ) I would like to sign my review report

English language and style

( ) Extensive editing of English language and style required

( ) Moderate English changes required

( ) English language and style are fine/minor spell check required

(x) I don't feel qualified to judge about the English language and style

Yes        Can be improved               Must be improved             Not applicable

Does the introduction provide sufficient background and include all relevant references?

(x)          ( )           ( )           ( )

Is the research design appropriate?

(x)          ( )           ( )           ( )

Are the methods adequately described?

(x)          ( )           ( )           ( )

Are the results clearly presented?

(x)          ( )           ( )           ( )

Are the conclusions supported by the results?

(x)          ( )           ( )           ( )

Comments and Suggestions for Authors

The manuscript describes the light-independent antimicrobial activity of synthetic cationic amphiphilic iron N-alkyl pyridylporphyrins. This is an interesting paper which is doubtlessly suitable for publication in “Antioxidants”. The paper is complete and well structured. The description of experimental procedure is detailed and results critically analyzed and commented. In summary, the measurements are adequate and the derived conclusions are consistent. Particularly interesting are results obtained for FeTnHex-2-Py that presents a strong bactericidal activity and lower toxicity to laboratory animals.

I think the manuscript can be accepted in this form and it needs no revisión.

Reviewer 3 Report

The scope of the manuscript was to investigate synthetic iron porphyrins as antimicrobial agents. The topic is timely and should attract potential readers. The experiments were well designed, the results were presented clearly and discussed comprehensively.

A few editorial notes:

  1. Latin names of the organisms should be italicized consistently throughout the manuscript. Please see lines: 41, 98, 141, 148, 168, 170, 180, 181, 199, 206, 218, 230, 260, 272, 278, 279, 280, 281, 284, 290, 292, 331, 332, 347, 348
  2. Line 126 'protein' should be replaced with 'protein concentration'.
  3. Authors should use either 'OD 600 nm' or 'OD600nm' throughout the manuscript.   Please see Lines 162 and 170 as an example of inconsistency.
  4. Authors should also replaced '(Figure X)' with '(Fig. X) while referencing figures at the end of sentences. Please see Line 178.

Author Response

Response to Reviewer 3

We are grateful for your help in improving the manuscript.

R3

Open Review

(x) I would not like to sign my review report

( ) I would like to sign my review report

English language and style

( ) Extensive editing of English language and style required

( ) Moderate English changes required

(x) English language and style are fine/minor spell check required

( ) I don't feel qualified to judge about the English language and style

Yes        Can be improved               Must be improved             Not applicable

Does the introduction provide sufficient background and include all relevant references?

(x)          ( )           ( )           ( )

Is the research design appropriate?

(x)          ( )           ( )           ( )

Are the methods adequately described?

(x)          ( )           ( )           ( )

Are the results clearly presented?

(x)          ( )           ( )           ( )

Are the conclusions supported by the results?

(x)          ( )           ( )           ( )

Comments and Suggestions for Authors

The scope of the manuscript was to investigate synthetic iron porphyrins as antimicrobial agents. The topic is timely and should attract potential readers. The experiments were well designed, the results were presented clearly and discussed comprehensively.

A few editorial notes:

Latin names of the organisms should be italicized consistently throughout the manuscript. Please see lines: 41, 98, 141, 148, 168, 170, 180, 181, 199, 206, 218, 230, 260, 272, 278, 279, 280, 281, 284, 290, 292, 331, 332, 347, 348

  • Formatting was lost when transferring to the template and this was overlooked. All are corrected.
  • Line 126 'protein' should be replaced with 'protein concentration'.
  • Corrected-           Corrected
    • Authors should also replaced '(Figure X)' with '(Fig. X) while referencing figures at the end of sentences. Please see Line 178.
    • Authors should use either 'OD 600 nm' or 'OD600nm' throughout the manuscript.   Please see Lines 162 and 170 as an example of inconsistency.
  • The template and Instructions to Authors state that “All figures and tables should be cited in the main text as Figure 1, Table 1, etc.” Because in papers already published in Antioxidants, “Figure” was not abbreviated, we decided not to abbreviate.
